# Upregulation of miR-33 Exacerbates Heat-Stress-Induced Apoptosis in Granulosa Cell and Follicular Atresia of Nile Tilapia (*Oreochromis niloticus*) by Targeting TGFβ1I1

**DOI:** 10.3390/genes13061009

**Published:** 2022-06-02

**Authors:** Jun Qiang, Fan-Yi Tao, Qi-Si Lu, Jie He, Pao Xu

**Affiliations:** 1Key Laboratory of Freshwater Fisheries and Germplasm Resources Utilization, Freshwater Fisheries Research Center, Chinese Academy of Fishery Sciences, Ministry of Agriculture, Wuxi 214081, China; taoyifan@ffrc.cn (F.-Y.T.); lusiqi@ffrc.cn (Q.-S.L.); hej@ffrc.cn (J.H.); xup@ffrc.cn (P.X.); 2Wuxi Fisheries College, Nanjing Agricultural University, Wuxi 214081, China

**Keywords:** *Oreochromis niloticus*, transforming growth factor-β1-induced transcript 1, miR-33, follicular development, apoptosis

## Abstract

High temperature affects egg quality and increases follicular atresia in teleosts. The present study aimed to explore the regulated mechanism of ovary syndrome of Nile tilapia (*Oreochromis niloticus*) exposed to heat stress. To this end, we conducted histological and biochemical analyses and integrated miRNA-target gene analyses. The histochemical analyses confirmed that heat stress promoted the apoptosis of granulosa cell and therefore resulted in increased follicular atresia in the ovary. Heat stress led to the differential expression of multiple miRNAs (miR-27e, -27b-3p, -33, -34a -133a-5p, and -301b-5p). In a luciferase activity assay, miR-33 bound to the 3′-untranslated region (UTR) of the *TGFβ1I1* (transforming growth factor-β1-induced transcript 1) gene and inhibited its expression. A *TGFβ1I1* gene signal was detected in the granulosa cells of Nile tilapia by immunohistochemical analysis. Up-regulation of the miR-33 of tilapia at 6 d and 12 d exposed to heat (34.5 °C ± 0.5 °C) had significant down-regulation of the *TGFβ1I1* expression of the gene and protein in tilapia ovaries. An miRNA-target gene integrated analysis revealed that miR-33 and TGFβ1I1 function in an apoptosis-related signal pathway. The signal transduction of the vascular endothelial growth factor (VEGF) family members VEGFA and its receptor (KDR) in the heat-stressed group decreased significantly compared with the control group. Transcript-levels of the *Bax* and *Caspase-3* as apoptotic promotors were activated and *Bcl-2* and *Caspase-8* as apoptotic inhibitors were suppressed in the heat-stressed tilapia. These results suggest that heat stress increases the expression of miR-33, which targets *TGFβ1I1* and inhibits its expression, resulting in decreased levels of follicle-stimulating hormone and 17β-estradiol and increased apoptosis by suppressing VEGF signaling, eventually inducing follicular atresia. In conclusion, our results show that the miR-33/TGFβ1I1 axis of Nile tilapia is involved in the follicular development of broodstock, and can suppress VEGF signaling to accelerate follicular atresia. Our findings demonstrate the suppressive role of miR-33 during oocyte development in Nile tilapia.

## 1. Introduction

The aquaculture industry requires large amounts of fish fry. The quality of fish eggs directly affects the fertilization and hatching rates after combining with sperm, which, in turn, affects the survival and quality of the fry [1,2]. Egg production is affected by genetic factors and by the size and age of the broodstock [3,4]. Most follicles do not develop properly and degenerate, in a process known as follicular atresia [5]. Many exogenous factors affect the rate of follicular atresia, such as food availability, pH, light, temperature, and stocking density [6,7,8,9]. For the welfare and sustainable development of aquatic animals, therefore, it is very important for us to understand the response by which stresses induce follicular atresia and to develop novel therapeutic targets.

Stress affects the rate of follicular atresia, likely because stress hormones (mainly epinephrine and cortisol) inhibit the reproductive hormone axis, which, in turn, restricts the hormone release (follicle-stimulating hormone, FSH; luteinizing hormone, LH) [10]. Acute stress can inhibit steroidogenesis in fish gonads, and has been introduced to damage the quality of fish egg such as rainbow trout (*Oncorhynchus mykiss*) [11,12]. Another study showed that crowding stress resulted in increased plasma cortisol and corticotropin hormones in broodstock, and decreased plasma testosterone and 17β-estradiol (E_2_) levels [10]. Fishing and crowding stresses were found to increase the plasma cortisol content, decrease E_2_ levels, and increase the proportion of atretic follicles during vitellogenesis in red gurnard (*Chelidonichthys kumu*) [13]. In white sturgeon (*Acipenser transmontanus* Richardson) [14] and rainbow trout [15], decreased serum E_2_ contents during spawning resulted in ovarian arrest and follicular atresia.

Members of the transforming growth factor-β (TGFβ) superfamily regulate various paracrine/autocrine pathways in the pituitary and ovary. These pathways mainly play an important role in the regulation of steroid hormones and cell proliferation and differentiation, gonadotropin receptor expression, and follicle selection and ovulation [16]. In the presence of FSH, TGFβ1 is linked to the proliferation of mammalian follicular granulosa and increases the expression of LH receptors (LHR), the secretion of E_2_, and the production of follicular fluid [17]. In aquatic animals, TGFβ superfamily members are mainly involved in early embryonic development [18], immune regulation [19,20], and tissue regeneration [21]. The roles of members of the TGFβ superfamily in follicular development and atresia in fish are still unclear. Studies on zebrafish oocyte development have revealed multiple sites of TGFβ1 action including LHR, 20β-hydroxysteroid dehydrogenase, and membrane progestin receptor β [22,23,24]. It has been shown that TGFβ1 mediates gonadotropin- and 17α, 20β-dihydroxyprogesterone-induced maturation of zebrafish oocytes in dose-related and time-dependent effects [22].

An important hallmark of follicular atresia is increased apoptosis in the granulosa cells [25]. Vascular endothelial growth factor (VEGF) and its corresponding signaling pathways are involved in the initiation and regulation of various apoptosis programs that occur during processes such as angiogenesis [26], cancer cell proliferation and differentiation [27], and cell regeneration [28]. A previous study using an endothelial cell model showed that cross-talk between VEGF and TGFβ1 regulates apoptosis [29]. TGFβ1 promotes the expression of FGF-2 in endothelial cells, resulting in increased VEGF synthesis and alleviation of apoptosis [29]. In addition, miRNAs play a very important role in the regulation of granulosa cell apoptosis. Several miRNAs in mammals, such as miR-130a, miR-31, miR-20b, miR-143, and miR-92a, have been reported to be involved in the regulation of proliferation and apoptosis of granulosa cells, and synthesis and secretion of hormones [30,31,32]. 

There are few studies on the regulatory mechanism of granulosa cell apoptosis and follicular atresia in teleosts, and most have been based on tissue observations and have not revealed the mechanisms underlying these events. In recent years, the tilapia industry has entered a bottleneck with problems of germplasm degradation, disease, and declining fertility. These problems greatly threaten the healthy and sustainable development of the industry. Although the number of eggs (5000–12,000 eggs/fish) has not changed much, the absolute fecundity (300–1500 eggs/fish) has decreased significantly. In summer in particular (July to September), when the water temperature is higher than 34 °C, tilapia cannot lay eggs or the quality of eggs is poor, and fry survival is low [25]. Therefore, analyzing the molecular mechanism of disordered follicular development in tilapia under heat stress is of great significance for the development of therapeutic treatments and for the selection of new varieties with high fecundity. In this study, miRNA sequence analyses identified key regulators and signaling pathways in response to heat stress challenges of the tilapia ovary. Further analyses revealed the roles of a specific miRNA and its target gene in this process. These results shed light on how follicular development is regulated and identify biomarkers for the treatment of heat-stressed broodstock.

## 2. Materials and Methods

### 2.1. Experimental Fish

The experimental Nile tilapia were bred from the same family. When the fish reached 18 cm in length, 200 female fish with a neat tail pattern and good gonad development were selected and reared in four circular 6 m diameter tanks. Water temperature (27–29 °C) and dissolved oxygen (>5.5 mg/L) were monitored daily. The fish in each tank were fed twice daily (30% protein and 7% fat), and the rearing cycle was 120 days. The abdomens of female fish were gently squeezed to release large quantities of mature eggs. We selected 80 Nile tilapia that had completed their first ovulation within 2 days for this experiment.

### 2.2. Experimental Procedure

Female Nile tilapia selected as described above (average fish weight: 412.67 g ± 27.89 g) were randomly distributed into tanks (each 4 m^3^) at a density of 10 fish per tank, and each treatment (heat-stressed group: 34.5 °C ± 0.5 °C and control group: 27.5 °C ± 0.5 °C) had four replicates. The experimental temperature was maintained by a water temperature control system. Fish management and feeding were conducted as described in Section 2.1. 

### 2.3. Sampling

Fish were sampled at 6 d and 12 d after the first spawning. At each sampling time, three experimental fish randomly selected from each tank were tested (a total of 12 fish per treatment group). An amount of 200 mg/L of MS-222 (Sigma, St. Louis, MO, USA) was used for deep anesthesia in sampled fish. The ovaries were quickly and completely dissected and photographed. Sampling and fixation of ovarian tissue samples for histological and TUNEL staining experiments followed our previous method [9,25]. A small piece of the ovarian tissue was placed dry into a sterile tube and snap-frozen in liquid nitrogen, and then freeze-stored (at −80 °C) until use in miRNA sequencing and gene transcript level analyses. Blood samples were collected from the tail vein of another 12 fish from each experimental group. The blood samples were centrifuged at 4000× *g* for 10 min to obtain serum that was then frozen (−20 °C) until use in serum steroid hormone analysis.

### 2.4. Observation of Oocyte Development and Apoptosis

Hematoxylin-eosin (HE) staining: The fixed samples were dehydrated, cleared, embedded in paraffin, and serially cut into sections, conducted as described by Qiang et al. [9]. The sections were observed and photographed through a NIKON digital sight DS-FI2 imaging system (Nikon, Tokyo, Japan). 

Fluorescent TUNEL staining: DNase-free proteinase K (20 μg/mL) was added dropwise to the dewaxed and rehydrated sections, and then they were incubated in a humid chamber at 37 °C for 30 min. In a dark environment, 50 μL of prepared TUNEL reaction solution was added dropwise to the tissue sections, and they were then incubated at 37 °C for 60 min in a humidified box. The sections on slides with anti-fluorescence quenching liquid were analyzed through a NIKON DS-U3 imaging system. The excitation/emission wavelengths for 4′-6-diamidino-2-phenylindole (DAPI) staining were 330–380 nm and 420 nm, respectively, and for fluorescein isothiocyanate (FITC staining), they were 465–495 nm and 515–555 nm, respectively. The TUNEL kit was from Yanjin Biotechnology Inc. (Shanghai, China).

### 2.5. miRNA Sequencing and Validation

RNA extraction and library construction: One portion of ovarian tissue was removed from storage at −80 °C, thawed in an ice box, and then cut into small pieces. RNA extraction and RNA quality assessment were according to Qiang et al. [25]. Samples from four fish from each tank were constructed into one miRNA sequencing library. A total of 12 sequencing libraries were constructed: three 6 d control groups (6CG_1, 6CG _2, 6CG _3) vs. three 6 d heat-stressed groups (6HG_1, 6HG_2, and 6HG_3); and three 12 d control groups (12CG_1, 12CG _2, 12CG _3) vs. three 12 d heat-stressed groups (12HG_1, 12HG_2, and 12HG_3). Based on standard methods of the LC-BIO (Hangzhou, China), we constructed and sequenced the miRNA libraries using the Illumina HiSeq^TM^ 6000 platform with single-end sequencing (50 bp). 

The total reads obtained by HiSeq sequencing were screened, and FastQC was used to eliminate 3′-end adapters, insertion fragments, 5′-end contaminants, reads less than 18 nt long, and polyA reads. Sequence alignment was performed using Bowtie software. The reads aligned to the Nile tilapia genome (http://asia.ensembl.org/Oreochromis_niloticus/Info/Index, accessed on 2 May 2020) were searched against miRNAs of specific species in the miRbase 22.0 database (http://www.mirbase.org/, accessed on 2 May 2020) using BLAST, and were identified as known conserved miRNAs. 

Analysis of differentially expressed (DE) miRNAs and bioinformatics analyses: The expression levels of miRNAs were normalized based on the modified reads per million reads (RPM) value. Significant differences in miRNA levels in the 6CG vs. 6HG and 12CG vs. 12HG comparisons were detected by Student’s t-test. The criteria for DE miRNAs in 6CG vs. 6HG and 12CG vs. 12HG were *p* < 0.05 and fold-change ≥1.5 or ≤0.67. We used TargetScan 7.2 and miRanda v3.3a to predict the target genes of the DE miRNAs. 

Real-time qPCR validation of miRNA: The reverse transcription and quantitative RT-PCR of miRNAs were performed following Qiang et al. [33]. *U6* was used as the reference gene for calculating the relative expression levels of miRNA. We analyzed three replicates of each sample, and the negative control had no template cDNA. The amplification reaction was performed on a 7900HT Fast Real-Time PCR system (Applied Biosystems, Foster City, CA, USA). At the end of the amplification, the temperature was increased from 60 °C, and the dissolution curve was used to verify the specificity of the amplified product.

### 2.6. Validation of miRNA Target Genes

A partial 3′-untranslated region (3′-UTR) sequence (approximately 300 bp in length) containing the seed-binding region of Nile tilapia *TGFβ1I1* was synthesized and connected with the pmirGLO vector using T4 ligase. A wild-type pmirGLO-*TGFβ1I1* 3′-UTR vector was similarly constructed. According to the principle of binding between miRNAs and their target genes, a 3′-UTR mutated sequence of the target gene (GUGACGUCCAUGUAA mutated to AGGCAGGACACUUGG) was designed, and the mutant pmirGLO-*TGFβ1I1* 3′-UTR vector was constructed. The 3′-UTR sequence was synthesized by Genewiz, Inc. Then, pmirGLO and the mutant or wild-type pmirGLO-*TGFβ1I1* 3′-UTR vector were ligated with a dual-luciferase reporter vector (pRL vector, Promega, Madison, WI, USA), and the constructs were transformed into competent cells. The *TGFβ1I1* 3′-UTR wild-type (*TGFβ1I1* 3′-UTR) and *TGFβ1I1* 3′-UTR mutant (*TGFβ1I1* mUTR) recombinant dual-luciferase reporter vectors were sequenced. HEK293T cells were co-transfected with the *O. niloticus* miR-33 mimic (oni-miR-33) or oni-miR-33 negative control (oni-NC) and the *TGFβ1I1* mUTR, *TGFβ1I1* UTR, or pmirGLO dual-luciferase reporter vector. After 48 h, the medium was removed, the cell lysate PLB was fully lysed, and then LARII was added to detect the activity of firefly luciferase. The ratio of firefly/*Renilla* luciferase activity was conducted to determine the experimental results. The oni-miR-33 mimic and oni-NC were designed and synthesized by Ribo-Bio Co., Ltd. (Guangzhou, China).

### 2.7. Detection of FSH, LH, and E_2_ Contents

The serum FSH, LH, and E_2_ contents were determined by enzyme-linked immunosorbent assay (ELISA) kits [25]. Samples (40 µL of serum of each sample) were added to wells in the microplate. After the serum hormone in each sample bound to the antibody in the microplate, color was developed using a chromogenic solution and HRP. Three separate determinations were conducted for each serum sample. A Multiskan Spectrum microplate spectrophotometer (BioTek Eon, Winooski, VT, USA) was used to determine the absorbance at a given wavelength for each hormone. The concentrations of hormones in the samples were calculated using the linear regression equation of the standard curve. 

### 2.8. Western Blot (WB) Analyses

Immunized New Zealand White rabbits were used to produce polyclonal antibodies of the Nile tilapia β-actin protein and TGFβ1I1 protein. WB analysis was conducted as described elsewhere [34]. Appendix A shows amino acid sequence information of the polypeptide antigens of the TGFβ1I1 protein and β-actin protein.

### 2.9. Immunohistochemical Staining

Ovary tissue sections were deparaffinized with xylene and dehydrated in an ethanol gradient. Endogenous enzymes were eliminated by incubating the sections with 3% H_2_O_2_ at room temperature for 10 min; the liquid was then shaken off and the sections were washed three times with phosphate-buffered saline (PBS) (5 min/wash) before being boiled in 0.01 M trisodium citrate in a microwave oven for 3 min; and then heated for 20 min to retrieve hot antigens. The sections were removed and cooled to room temperature, then washed three times with PBS (5 min/wash). Goat serum was added and the sections were incubated in a sealed, humid box for 1 h at room temperature. The liquid was again shaken off, and the primary antibody (TGFβ1I1 antibody, diluted 1:200) was added; then, the sections were kept overnight at 4 °C. The following morning, the primary antibody was shaken off, and the sections were washed three times with PBS (10 min/wash). The secondary antibody (diluted concentration 1:400) was added and the sections were incubated in a humid box for 1 h at room temperature before shaking off the secondary antibody and washing three times with PBS (10 min/wash). Then, DAB color reagent was added drop by drop. The color was developed at room temperature for 5–10 min; then, the color reagent was shaken off and the sections were washed three times with ddH_2_O (5 min/wash). After HE counterstaining, the sections were dehydrated with absolute ethanol for 3 min, cleared in xylene for 3 min, allowed to dry, then sealed with neutral gum. For the negative control, the primary antibody was replaced with normal rabbit serum. The sections were observed and photographed under an Eclipse ci-L microscope (Nikon, Tokyo, Japan). 

### 2.10. Analysis of Gene Transcript Levels

The RT and qRT-PCR analyses of the gene mRNAs were performed as previously reported [25]. The β-actin gene was used as an internal reference in our study. Primers to amplify genes of interest and β-actin were synthesized by Genewiz, Inc (Appendix A). The 2^−ΔΔCt^ method was used to calculate the transcript levels of mRNA.

### 2.11. Data Analysis

The results are shown as mean plus standard deviation (mean ± SD). The experimental data were subjected to one-way analysis of variance, carried out via SPSS 25.0. First, the data were tested for a normal distribution and homogeneity of variance. Significant differences among different sampling times within the same experimental group were detected by the independent samples *t*-test. Significant differences between experimental groups at each sampling time were detected by the paired samples *t*-test. The significance level was *p* < 0.05. The very high significance level was *p* < 0.01.

## 3. Results

### 3.1. Morphological and Histological Observations of Nile Tilapia Ovaries at 6 d and 12 d of Heat Stress

We monitored the development of the ovary of Nile tilapia after the first spawning. At 6 d after spawning, the ovary was slender and orange-yellow (Figure 1A(1)). Histological analyses showed that oocytes were present in various phases, with most in the monolayer follicular stage (stage II) or vitellogenin stage (stage III). A layer of follicular cells and flat radial bands surrounded the phase III oocytes (Figure 1B(1)). Atretic follicles and yolk-filled oocytes (stage V) that did not degenerate during the experimental time period were also present in the ovary. Compared with the ovaries in the control fish at 6 d, those in heat-stressed fish at 6 d were significantly larger (*p* = 0.04 < 0.05; Table 1), and contained more white eggs (Figure 1A(2)) and more oocytes at early stage IV and stage III (Figure 1B(2)). Some stage IV oocytes showed atretic features. The follicular cell layer and zona radiata were isolated respectively from the oocytes. In addition, the gradually decomposing yolk material was shown in the cytoplasm of oocytes.

The ovaries of Nile tilapia in the control group at 12 d were full, and the eggs were dark yellow, clearly defined, and countable (Figure 1A(3)). The ovaries were larger at 12 d than at 6 d. Most oocytes in the ovary were at stage V, and the eggs were about to mature and be released (Figure 1B(3)). At 12 d in the heat-stressed group, the ovaries were white and many white eggs were visible to the naked eye (Figure 1A(4)). There were more atretic follicles in the ovaries of heat-stressed fish than in those of the control fish at 12 d (*p* = 0.03 < 0.05; Table 1). The ovaries were dominated by atretic follicles, the yolk material was completely decomposed, and many stage II and III oocytes were atretic (Figure 1B(4)). The degenerated follicles had only one cavity, which was gradually surrounded by surrounding connective tissue, and orange-brown pigment cells were present in some areas (Figure 1A(4)).

### 3.2. Apoptosis in Nile Tilapia Ovarian Granulosa Cells at 6 d and 12 d of Heat Stress

In TUNEL staining analyses, FITC fluorescein-labeled positive apoptotic nuclei are green. As shown in Figure 1C(1,3), in the control group of Nile tilapia (water temperature 27.5 °C), there were a few apoptotic cells observed in the granular cell layer that continued to develop for 6 and 12 d after the first spawning. The heat-stressed tilapia follicular had significantly more apoptotic granulosa cells at 6 d and 12 d compared to the control group (Figure 1C(2,4)); and the number of apoptotic cells increased during ovary development.

### 3.3. Serum FSH, LH, and E_2_ Contents of Nile Tilapia at 6 d and 12 d of Heat Stress

The serum FSH and E_2_ contents of Nile tilapia at 6 d were significantly lower in the heat-stressed group than in the control group (*p* = 0.03 < 0.05; Figure 2), and although the serum LH content was slightly lower than that of the control group, the difference was not significant (*p* = 0.07 > 0.05). At 12 d, the serum LH, FSH, and E_2_ contents were significantly higher in the control group than in the heat-stressed group (*p* = 0.03, 0.03, 0.02 < 0.05). 

### 3.4. DE miRNAs between Control and Heat-Stressed Nile Tilapi

The quality assessment of 6CGs vs. 6HGs and 12CGs vs. 12HGs is shown in Appendix A. The conservation and the length distribution of the valid reads in the six miRNA libraries are shown in Appendix A. We screened DE miRNAs in the 6CG vs. 6HG and 12CG vs. 12HG miRNA libraries. In the 6CG vs. 6HG comparison, 115 miRNAs were up-regulated and 76 miRNAs were down-regulated (|log2foldchange| =1 and *p* value (*t*-test) < 0.05) (Figure 3A). In the 12CG vs. 12HG comparison, 111 miRNAs were up-regulated and 111 miRNAs were down-regulated. To verify the results, we conducted qRT-PCR analyses of selected DE miRNAs from the two sets of sequencing results. Among them, miR-27e, miR-27b-3p, miR-33, and miR-34a were significantly up-regulated in HGs than in CGs at 6 d and 12 d, while miR-133a-5p and miR-301b-5p were significantly down-regulated in HGs than in CGs (Figure 3D). The qRT-PCR results were consistent with the sequencing results (Figure 3B,C), confirming their reliability. In this experiment, we focused on the potential regulatory pathway of miR-33 in the ovary of heat-stressed Nile tilapia. 

### 3.5. Analysis of miR-33 Sequence

Our results revealed that the tilapia pre-mir-33 (oni-mir-33) consisted of 88 nt containing a 19-nt mature sequence (on-miR-33). A secondary structure analysis of oni-mir-33 at the mfold Web Server revealed a well-developed hairpin structure in its sequence (Figure 4A), and confirmed that it is conserved among multiple species (Figure 4B). Using tools at the miRBase database (https://www.mirbase.org/, accessed on 20 May 2020), we screened the mir-33 sequences of multiple species and constructed a phylogenetic tree using oni-mir-33 sequences showing a high sequence identity with ola-mir-33 from the model animal medaka (*Oryzias latipes*). As shown in Figure 4C, there were two distinct clades in the oni-mir-33 phylogenetic tree: vertebrates and invertebrates.

### 3.6. Enrichment Analysis to Identify Potential Target Genes of miR-33

We preliminarily predicted the potential target genes of oni-miR-33 using TargetScan 7.2 and miRanda v3.3a, and then conducted Kyoto Encyclopedia of Genes and Genomes (KEGG) analyses to explore their potential functions. In the 6CG vs. 6HG comparison (Figure 5A), the potential target genes of oni-miR-33 were involved in multiple cell fate control, oocyte development and maturation, metabolism, and hormone regulation. In the 12CG vs. 12HG comparison (Figure 5B), the potential target genes of oni-miR-33 were involved in cell fate control, oocyte development and maturation, metabolism, and hormone regulation. The enrichment of several signaling pathways with targets of oni-miR-33 was detected at both 6 d and 12 d. The VEGF signaling pathway, which is involved the control of cell fate, was selected for further analysis. 

### 3.7. miR-33 Directly Regulates the Expression of TGFβ1I1

The bioinformatics analysis revealed that oni-miR-33 directly targets the 3′-UTR of the *TGFβ1I1* gene in Nile tilapia (Figure 6A). The minimum free energy of oni-miR-33 and the target gene *TGFβ1I1* was −21.0 kcal/mol. We cloned the Nile tilapia *TGFβ1I1* 3′-UTR into a luciferase reporter plasmid (Figure 6B), and then oni-miR-33 and pmirGLO, pmirGLO-TGFβ1I1 3′-UTR, or pmirGLO-TGFβ1I1 mUTR were co-transfected into HEK293T cells. Compared with the other treatment groups, the cells co-transfected with oni-miR-33 and mirGLO-TGFβ1I1 3′-UTR showed a significantly lower luciferase activity (*p* = 0.03 < 0.05; Figure 6C), confirming that miR-33 specifically targets the 3′-UTR of the *TGFβ1I1* gene in Nile tilapia. 

### 3.8. miR-33 Participates in Apoptosis of Nile Tilapia Follicle Granulosa Cells through VEGF Signaling

The results of qRT-PCR (Figure 7A) and Western blotting (Figure 7B) analyses showed that the *TGFβ1I1* transcript and TGFβ1I1 protein levels in the ovary of Nile tilapia were significantly lower in the heat-stressed group than in the control group at 6 d and 12 d. The TGFβ1I1 protein was detected in the granulosa cell layer of Nile tilapia follicles by immunohistochemical analysis (Figure 7C), and the positive signal in the granulosa cell layer of the control group at 12 d was significantly stronger than that of 6 d (Figure 7C(1,3)). A weaker positive signal was detected in the granulosa cells detached from the degenerated follicle periphery at 12 d in the heat-stressed group (Figure 7C(4)). Next, we determined the transcript levels of key genes involved in the VEGF signaling pathway and apoptosis (Figure 8). The transcript levels of *VEGFA* (encoding vascular endothelial growth factor A) and *KDR* (encoding vascular endothelial growth factor receptor II) in the Nile tilapia ovary were significantly lower in the heat-stressed group than in the control group 6 d and 12 d (*p* = 0.04, 0.03 < 0.05); but the *VEGFB* levels had no significant difference between the heat-stressed group and the control group (*p* = 0.12 > 0.05). The *Bax* and *Caspase-3* levels at 6 d and 12 d were significantly higher in the heat-stressed group than in the control group (*p* = 0.04, 0.04, 0.03, 0.04 < 0.05), while the transcript levels of *Bcl-2* were significantly lower in the heat-stressed group than in the control group at 6 d and 12 d (*p* < 0.05). The transcript level of *Caspase-8* was significantly lower in the heat-stressed group than in the control group at 12 d (*p* = 0.03 < 0.05), but not at 6 d (*p* = 0.14 > 0.05). 

## 4. Discussion

### 4.1. miRNA Profiles Reveal Multiple Key miRNAs Involved in Development and Atresia

Follicular atresia is a complex biological process that limits the potential reproductive capacity of fish. Most previous studies on the specific roles of miRNAs in regulating follicular atresia have focused on mammals, so there is still much to learn about their roles in teleosts. The regulation of granulosa cell survival and death is critical for determining follicle fate [35]. Previous studies have shown that various miRNAs are involved in the regulation of granulosa cell apoptosis. For example, miR-17-5p regulates the function of porcine granulosa cells by targeting E2F1 (encoding E2F transcription factor 1) [36]. Other studies have shown that the overexpression of miR-17-5p significantly reduces E2F1 activity, while E2F1 knockdown promotes porcine granulosa cell growth and upregulates marker genes of follicle development, such as those encoding the LH receptor, cytochrome P450 19A1, and amphiregulin, thereby promoting E_2_ synthesis [36,37]. A recent study reported that miR-31 directly binds to the target genes HSD17B14 and FSHR to maintain the balance between proliferation and apoptosis of porcine ovarian granulosa cells and regulate steroid hormone metabolism [32].

In the present study, our data indicated that the differentially expressed miRNAs identified may act as dominant regulators in follicular development and atresia of female fish during heat stress, such as miR-27e, miR-27b-3p, miR-33, miR-34a, miR-133a-5p, and miR-301b-5p. Another member of this family of miRNAs, miR-34c, promotes apoptosis and inhibits the proliferation of porcine granulosa cells by targeting the forkhead box O3a gene [38]. In human granulosa cells, miR-23a and miR-27a target SMAD5 and regulate apoptosis through the FasL-Fas pathway. The down-regulation of SMAD5 results in an increased abundance of Fas, FasL, and cleaved Caspase-8 and Caspase-3 proteins, thereby promoting granulosa cell apoptosis [39]. Creb1 was identified as a direct and functional target of miR-27a-3p. A study on polycystic ovary syndrome using a mouse model revealed that the enhanced expression of miR-27a-3p significantly inhibits granulosa cell proliferation and promotes apoptosis by targeting Creb1 [40]. Therefore, these DE miRNAs detected in the present study may have important roles in the initiation of follicular atresia in tilapia. In this study, we focused on the role of miR-33 and its target gene TGFβ1I1 in the follicular development and atresia. 

### 4.2. miR-33 Directly Interacts with the 3′-UTR of TGFβ1I1

In this study, miR-33 was identified as a novel miRNA targeting Nile tilapia *TGFβ1I1*. We found that miR-33 inhibited the expression of *TGFβ1I1* in ovaries by directly binding to its 3′-UTR. The up-regulation of miR-33 under heat stress decreased the level of TGFβ1I1 protein. Previous studies have shown that miRNAs can affect mRNA stability by repressing translation as a result of their complementary binding to the 3′-UTR of mRNAs [41]. Our analyses showed that oni-miR-33/mir-33 shares a high sequence similarity with those in other organisms, suggesting that miR-33 is highly conserved and may be a key regulator. In fact, miR-33 has been reported to be involved in age-related macular degeneration [42], inflammatory responses [43], and cell proliferation in mice [44]. The differential expression of miR-33 in tilapia ovarian tissue under heat stress provides new clues about its role in regulating follicular atresia.

TGFβ1 is a multifunctional growth factor involved in granulosa cell growth, differentiation [45], proliferation [14], and apoptosis via multiple signaling pathways or related target genes [46]. As a pro-apoptotic factor, miR-130a markedly suppresses the expression of target gene *TGFβ1* and promotes apoptosis of porcine granulosa cells [47]. miR-126 attenuates the NORFA-induced TGFβ signaling pathway in ovarian granulosa cells by targeting *TGFβ2* expression, thereby inducing follicular atresia triggered by apoptosis in granulosa cells [31]. In the present study, we focused on *TGFβ1I1* as a target gene of the TGFβ signaling pathway. The up-regulation of miR-33 under heat stress may act as a pro-apoptotic factor, directly targeting *TGFβ1I1* to increase the apoptosis of granulosa cells and accelerate follicular atresia in Nile tilapia.

### 4.3. miR-33 Interferes with Steroid Hormone Synthesis by Targeting TGFβ1I1

TGFβ1 is involved in follicle growth and development. It affects FSH to regulate ovarian granulosa cell proliferation and differentiation, and its signal transduction-related abnormalities may lead to infertility [48]. In this study, the serum FSH and E_2_ contents in Nile tilapia were markedly suppressed in the heat-stressed group at 6 d and 12 d. The up-regulation of miR-33 may contribute to the inhibition of steroid hormone synthesis by directly targeting *TGFβ1I1*, leading to increased follicular atresia. Previous studies have found that TGFβ1/2 are mainly expressed in theca cells, and TGFβ1 can increase the expression of VEGF in granulosa cells and promote angiogenesis [49]. As a downstream target gene of *TGFβ1*, *TGFβ1I1* is mainly expressed in the granulosa cell layer of Nile tilapia follicles, especially in the later stages of oocyte development. The follicular cell layer is an important synthesis site for hormones, nutrients, and other signaling molecules [50]. These substances are transported to oocytes via microvilli and blood vessels to regulate oocyte development and provide nutrition for oocyte growth [51]. The expression of TGFβ1I1 in granulosa cells of tilapia was significantly attenuated under heat stress. The down-regulation of *TGFβ1I1* transcript and TGFβ1I1 protein levels may have inhibited the development of blood vessels and the synthesis of steroid hormone receptors. If so, this would restrict the transport of hormones and nutrients, leading to an increase in follicular atresia. Therefore, TGFβ1I1 may play an important role in regulating the development and maturation of fish follicles. In the mouse model, in vitro fertilization of the egg obtained by injecting TGFβ1 into mouse ovarian cysts inhibits oocyte development, possibly because TGFβ1 induces the premature differentiation and proliferation of granulosa cells. This results in a large amount of follicular fluid and a rapid increase in follicle volume, so that oocytes are excreted before they mature [17]. Whether TGFβ1 induces TGFβ1I1 to regulate follicle development is still unknown. However, the significant up-regulation of TGFβ1I1 at 12 d in the control group helps to increase the FSH, E_2_, and LH hormone release, and accelerate follicular maturation.

### 4.4. miR-33 Targets TGFβ1I1 to Mediate VEGF Signaling, Contributing to the Regulation of Granulosa Cell Apoptosis

The ovary is an important site for animal oogenesis, follicle development, and corpus luteum formation and degeneration. Blood vessels transport materials to the ovaries, and this ensures that they can fulfil their role in reproductive physiology. A rich vascular network is conducive to the transportation of nutrients, so that follicles are able to develop well. If the vascular network forms poorly or becomes compromised, it will cause follicular atresia. The results of the KEGG analysis indicated that *TGFβ1I1* is involved in the regulation of the VEGF signaling pathway. VEGFA, a gene encoding a heparin-binding protein, plays an important role in angiogenesis by inducing the proliferation and migration of vascular endothelial cells [52]. VEGF, as an angiogenic factor, binds to corresponding receptors on the follicular granulosa cell membrane to trigger downstream signaling cascades and promote follicle development [53,54]. A previous study indicated that the Lentivirus-mediated overexpression of VEGFA in mouse granulosa cells significantly decreases the expression of *Bax* and *Caspase-3*, thereby inhibiting granulosa cell apoptosis [55]. The direct addition of VEGFA to mammalian ovary granulosa cells was shown to promote cell proliferation and inhibit apoptosis and follicular atresia [56]. In the present study, TGFβ1I1 as a potential inhibitor of apoptosis, and the significant inhibition of TGFβ1I1 down-regulated the expression of *VEGFA* (but not *VEGFB*) in granulosa cells when Nile tilapia were exposed to heat stress. This may have hindered angiogenesis so that the transport of hormones and nutrients was impaired, leading to increasing apoptosis (as demonstrated by increased transcript levels of *Bax* and *Caspase-3*). VEGFA affects oxygen supply during follicle development. In the mouse model, the direct injection of VEGFA into the ovaries was shown to promote angiogenesis and increase the number of follicles [57]. KDR is the main functional receptor of VEGF and is present only in vascular endothelial cells. The binding of KDR to VEGF activates endothelial cell division and migration [29]. Therefore, it is likely that heat stress not only inhibited angiogenesis in Nile tilapia, but also weakened the binding of VEGF to its receptors, thereby inhibiting the proliferation of follicular endothelial cells, resulting in an insufficient oxygen supply.

From our study, the heat-stressed group indicated higher expression levels of miR-33 in the ovaries at 6 d and 12 d, and lower levels of *TGFβ1I1*, *VEGFA*, and *KDR*. The heat-stressed group also showed increased expression levels of *Bax* and *Caspase-3*, and decreased levels of *Bcl-2* and *Caspase-8*. On the basis of these findings, combined with the results of histological analyses of oocytes in the heat-stressed group, we speculate that miR-33 targets *TGFβ1I1* to negatively regulate VEGF signaling, thereby promoting apoptosis in granulosa cells and resulting in follicular atresia. The bioinformatics analysis revealed that miR-33 has multiple potential targets including *HSP90AB1*, *PIK3R5*, *SMC1Al*, and *COL4A4*. The genes regulated by miRNAs may be useful targets for manipulation to improve the productive and reproductive performance traits of tilapia in the future.

## 5. Conclusions

From our study, we identified that miR-33 directly targets *TGFβ1I1*, which resulted in down-regulated levels of TGFβ1I1 transcripts and the TGFβ1I1 protein in the ovary of heat-stressed Nile tilapia. Our results show that the miR-33/TGFβ1I1 axis plays a major role in determining follicular granulosa cell apoptosis in Nile tilapia. Our results also show that miR-33, as a pro-apoptotic factor, suppresses VEGF signaling to accelerate follicular atresia, indicative of the role of the miR-33/TGFβ1I1 axis in follicle development and maturation of broodstock. Therefore, miR-33 is a non-hormonal regulator of follicular atresia. The target genes of miR-33 are useful candidates as molecular biomarkers for the tilapia breeding to improve reproductive performance.

## Figures and Tables

**Figure 1 genes-13-01009-f001:**
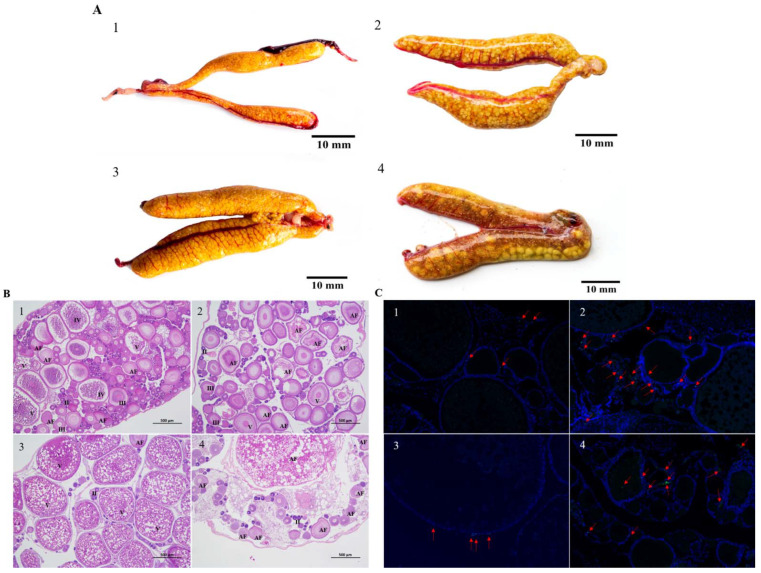
The gonadal status of Nile tilapia (*Oreochromis niloticus*) under heat stress. Note: (**A**) 1–4: Morphological changes of Nile tilapia ovary. (**B**) 1–4: Oocyte development of Nile tilapia ovary (magnification ×40, scale bar: 500 μm). II, III, IV, and V: stages II, III, IV, and V oocytes, respectively; AF: atretic follicles. Stage II oocytes, early perinucleolar oocytes; stage III oocytes, late perinucleolar oocytes; stage IV oocytes, vitel-logenic oocytes; stage V oocytes, maturing oocytes. (**C**) 1–4: Granulosa cell apoptosis of Nile tilapia ovary (magnification ×100, scale bar: 200 μm). Red arrows indicate apoptotic granulosa cells. a and c correspond to the control group (CG) on days 6 and 12, respectively; b and d correspond to the heat-stressed group (HG) on days 6 and 12, respectively.

**Figure 2 genes-13-01009-f002:**
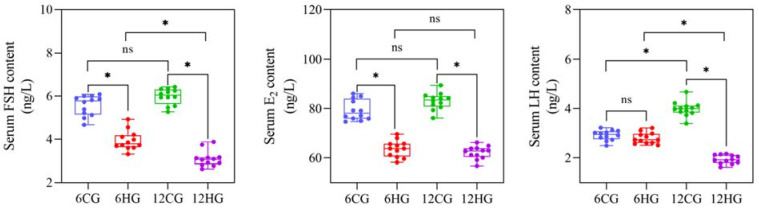
Serum FSH, E_2_, and LH contents in Nile tilapia (*Oreochromis niloticus*) under heat stress (*n* = 12). Note: * indicates statistical significance between heat-stressed group and control group (*p* < 0.05). ns indicates no statistical significance.

**Figure 3 genes-13-01009-f003:**
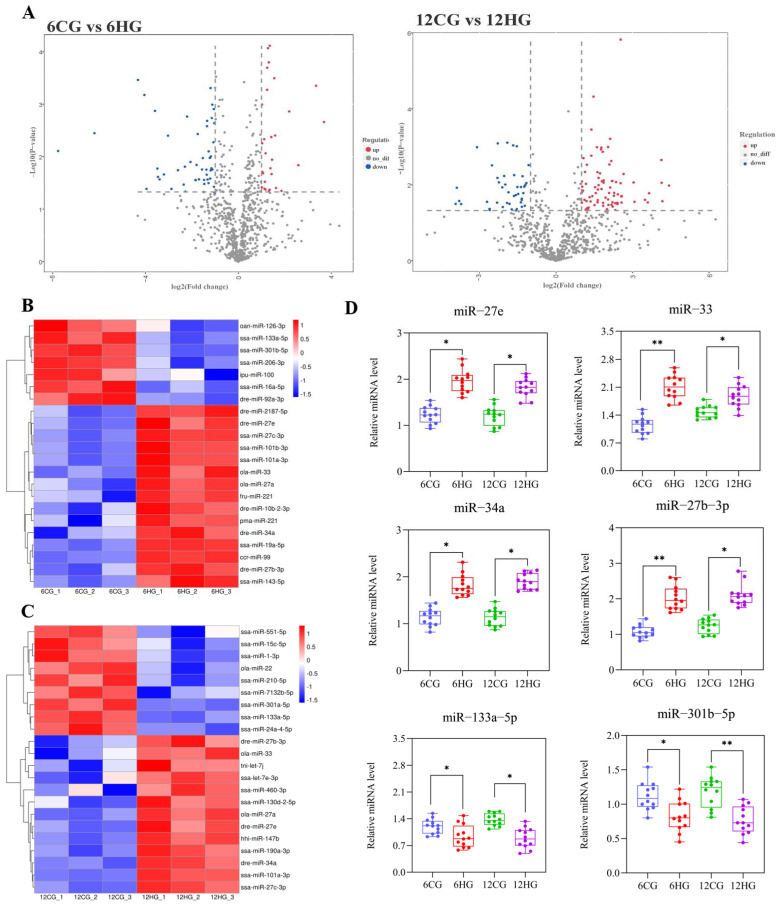
Differentially expressed miRNA screening and validation of Nile tilapia (*Oreochromis niloticus*) ovary under heat stress. Note: (**A**) Number of differentially expressed miRNAs. RNA from four samples was mixed to construct each sequencing library: 12 sequencing libraries were constructed, three 6 d control groups (6CG_1, 6CG _2, 6CG _3) vs. three 6 d heat-stressed groups (6HG_1, 6HG_2, and 6HG_3), and three 12 d control groups (12CG_1, 12CG _2, 12CG _3) vs. three 12 d heat-stressed groups (12HG_1, 12HG_2, and 12HG_3). (**B**,**C**) Heatmap analysis of differentially expressed miRNAs identified from miRNA-Seq data. (**D**) Validation of differentially expressed miRNAs by qRT-PCR (*n* = 12). * indicates statistical significance between heat-stressed group and control group (*p* < 0.05). ** indicates a statistically very significant difference between heat-stressed group and control group (*p* < 0.01). ns indicates no statistical significance.

**Figure 4 genes-13-01009-f004:**
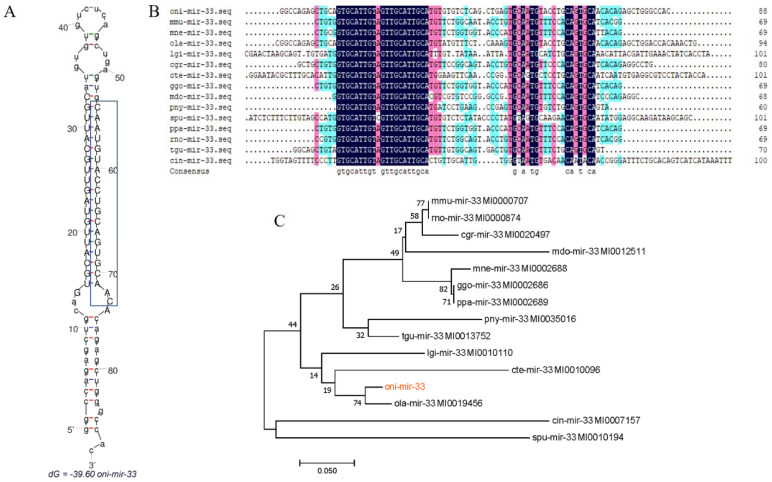
Characteristics of miR-33. (**A**) Hairpin structures of oni-mir-33 from *Oreochromis niloticus*. Blue box shows mature miR-33 sequence (oni-miR-33). (**B**) Analysis of Pre-mir-33 sequence by Multiple Sequences Alignment. (**C**) The phylogenetic tree was constructed by the neighbor-joining method using MEGA 7.

**Figure 5 genes-13-01009-f005:**
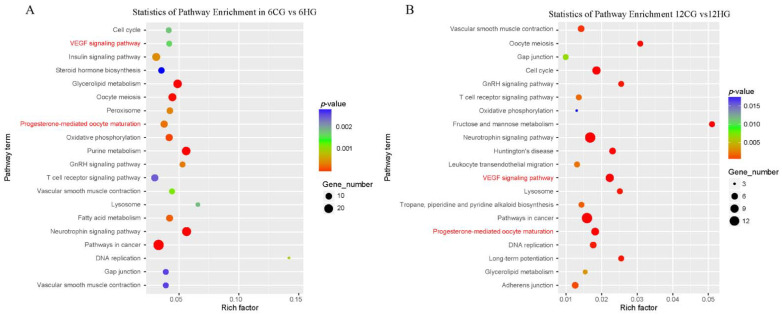
Analysis of enrichment pathways of miR-33 potential target gene (**A**,**B**). Note: KEGG subclasses enriched with target genes of miR-33, and signal pathways of potential target genes of miR-33. Circle diameter indicates quantity of potential target genes in the corresponding pathway.

**Figure 6 genes-13-01009-f006:**
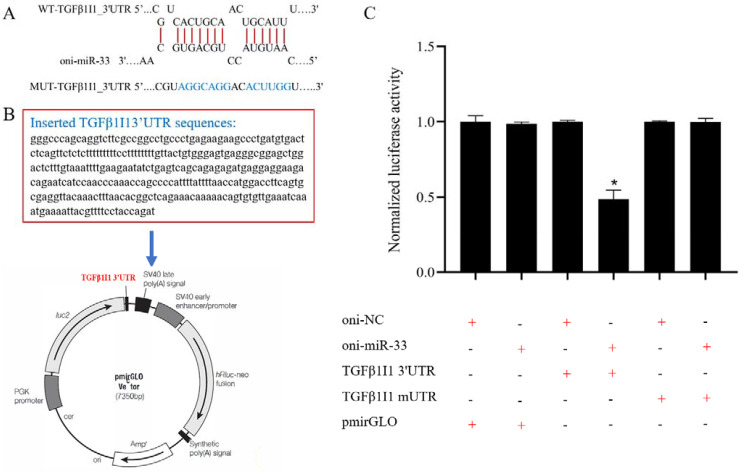
Identification of oni-miR-33 target gene *TGFβ1I1*. Note: (**A**) Schematic diagram of miR-33 binding to 3′-untranslated region (UTR) of *TGFβ1I1*, and wild or mutated UTR (mUTR) of tilapia *TGFβ1I1.* (**B**) Plasmid containing 3′-UTR of *TGFβ1I1* used for luciferase reporter analysis; (**C**) Luciferase activity in cells co-transfected with the *TGFβ1I1* 3′-UTR luciferase reporter plasmid and oni-miR-33 and pmirGLO, the pmirGLO-*TGFβ1I1* 3′-UTR, or pmirGLO-*TGFβ1I1* mUTR. * indicates statistical significance among different treatment groups (*p* < 0.05).

**Figure 7 genes-13-01009-f007:**
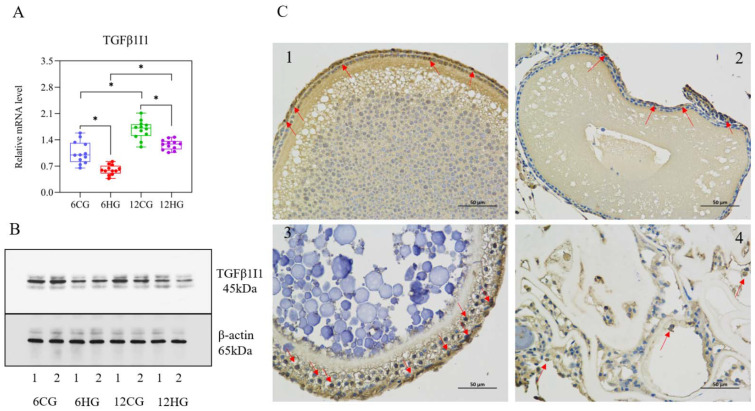
*TGFβ1I1* transcript and TGFβ1I1 protein levels, and distribution of TGFβ1I1 protein of Nile tilapia (*Oreochromis niloticus*) ovary under heat stress. Note: (**A**) *n* = 12. * indicates statistical significance between heat-stressed group and control group (*p* < 0.05). (**B**) Representative images of Western blot from female fish in 6CG, 6HG, 12CG, and 12HG. (**C**) 1–4: Red arrow indicates positive expression of TGFβ1I1 protein (scale bar: 50 μm). Red arrows indicate positive signal of TGFβ1I1 protein. A and C correspond to control group (CG) on days 6 and 12, respectively; B and D correspond to heat-stressed group (HG) on days 6 and 12, respectively.

**Figure 8 genes-13-01009-f008:**
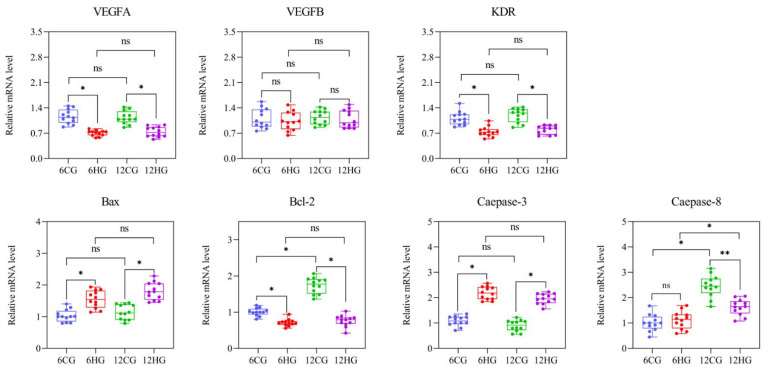
Activation of VEGF signaling and apoptosis contribute to follicular atresia of Nile tilapia (*Oreochromis niloticus*) ovary under heat stress. Note: *n* = 12. * indicates statistical significance between heat-stressed group and control group (*p* < 0.05). ** indicates statistically very significant difference between heat-stressed group and control group (*p* < 0.01). ns indicates no statistical significance.

**Table 1 genes-13-01009-t001:** The gonadal status both heat-stressed group and control group.

Measurement		
Sample at 6 d	Control group	Heat-stressed group
Gonadal weight (g)	8.76 ± 1.23 ^b^	10.56 ± 1.79 ^a^
The number of atretic oocytes	48–69	72–108
Mean of atretic follicles	58 ^b^	90 ^a^
Sample at 12 d		
Gonadal weight (g)	12.05 ± 2.21 ^a^	6.11 ± 1.84 ^b^
The number of atretic oocytes	44–78	84–122
Mean of atretic follicles	61 ^b^	103 ^a^

Note: Identification and counting of atretic follicles (AF) followed Qiang et al. [25]. The main features of AF are zona radiata fragmentation, nuclear disarrangement, and yolk granule breakdown and reabsorption. Different lowercase letters show significant differences between experimental groups (*p* < 0.05)

## Data Availability

The data used to support the findings of this study are included within the manuscript.

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
