# Peer review of "Upregulation of miR-33 Exacerbates Heat-Stress-Induced Apoptosis in Granulosa Cell and Follicular Atresia of Nile Tilapia (Oreochromis niloticus) by Targeting TGFβ1I1"

_genes, 2022, doi:10.3390/genes13061009_

Round 1

Reviewer 1 Report

This is a great aritcle on the effect of stress upon atresia in Tilapia. It reads well but has a bit of a jump from the general description of the phenotype to the miRNA results and the dialing in on one single miRNA. I have found another articloe of the authors and am not entirely sure about the overlap as I cannot read the other article as I do not have access so please excuse my hesitation. 

1) The authors have published a paper in Aquaculture (to which I sadly do not have access). In this the authors "analyzed tissue sections, serum hormone contents, gene transcript levels" and "showed significantly reduced gonadal weight and gonadosomatic index of female fish". The authors show the same results for this paper Figure 1, 2, 4 and Table 1. How is this data different? From the abstract it feels a little (and please excuse me if I am wrong) as if the authors may (!!!) have done a study with descriptive aspects as well as sequencing (RNAseq, miRNA seq) and written the seq results into two papers partially reusing the descriptive aspects. Is it a different cohort? Is there something critically different and/or am I reading the Aquaculture abstract not correctly? Would it be possible to submit a copy of the Aquaculture paper in order to see what of this data if any is duplicated? And if this data is duplicated, could the authors please remove the relevant parts from this paper and reference their previous work where appropriate?

2) Heat stress results in reduced serum LH/FSH, is there a known pathway that mitigates the stress effect upon these pituitary genes centrally? In the discussion the authors allude to the reduction in plasma FSH/LH top be due to local effects of TGFb (section 4.3) - this section is very tentative and to be fair I think requires some more discussion of central pathways since the main source of FSH/LH is generally thought to be the pituitary and not the oocyte. For this conclusion additional data towards the central expression of LH/FSH after heat show, i.e. pituitary expression after heat shock in this paradigm would be ciritcal! Maybe this is explained in the authors other publication?

3) There is a logical jump between the descriptive work and the sequencing results and the jump from the DE miRNAs to miR-33 is not clear to me. What about the other miRNAs? Could the authors perform some pathway analysis to see whether there are common aspects between the upregulated miRNA and between the downregulated miRNAs? What about 33 vs 34a? Or 27e vs 27b? I appreciate tha the authors found interesting resutls with miR33, but some form of analysis of the remainder of the miRNAs would be well placed here if only to provide a reference for a future paper of the authors looking at another one of these targets.

Minor comments:

Line 324 - you switch the order in which you express, siognificantly higher and significantly lower meaning the same since you flipped the groups. This could be confusing for the reader and it might be more easily understandable if you keep the order the same in both sentences.

Line 355 - was the qPCR carried out on the same samples as the sequencing or on a different cohort?

Data for transcript levels is expressed as % of b actin or % of U6. In the methods the authors mention the ddCT method. The results of that method are not in % of control gene. Could you please expand how this data was calculated or correct the legend?

Author Response

Comment 1: The authors have published a paper in Aquaculture (to which I sadly do not have access). In this the authors "analyzed tissue sections, serum hormone contents, gene transcript levels" and "showed significantly reduced gonadal weight and gonadosomatic index of female fish". The authors show the same results for this paper Figure 1, 2, 4 and Table 1. How is this data different? From the abstract it feels a little (and please excuse me if I am wrong) as if the authors may (!!!) have done a study with descriptive aspects as well as sequencing (RNAseq, miRNA seq) and written the seq results into two papers partially reusing the descriptive aspects. Is it a different cohort? Is there something critically different and/or am I reading the Aquaculture abstract not correctly? Would it be possible to submit a copy of the Aquaculture paper in order to see what of this data if any is duplicated? And if this data is duplicated, could the authors please remove the relevant parts from this paper and reference their previous work where appropriate?”

Response: Thank you for your careful review. The published manuscript is completely different from this manuscript. The manuscript published in Aquaculture mainly studies the effect of different treatment time (0 d, 2 d, 4 d, 8 d and 16 d) on the ovarian development of Nile tilapia under heat stress. Although some measures were the same (gene expression and serum hormones), the published manuscript focused on transcriptome analysis at 8 days of stress. This study is a completely new experiment, mainly to study the changes of miRNA omics during 6 d and 12 d of high temperature stress, and to reveal the molecular mechanism of follicular atresia through the regulation of miR-33 and its target genes. Please check the data and figures of the manuscript, there is no similarity, and there is no problem of multiple submissions for one manuscript.

Comment 2: Heat stress results in reduced serum LH/FSH, is there a known pathway that mitigates the stress effect upon these pituitary genes centrally? In the discussion the authors allude to the reduction in plasma FSH/LH top be due to local effects of TGFb (section 4.3) - this section is very tentative and to be fair I think requires some more discussion of central pathways since the main source of FSH/LH is generally thought to be the pituitary and not the oocyte. For this conclusion additional data towards the central expression of LH/FSH after heat show, i.e. pituitary expression after heat shock in this paradigm would be ciritcal! Maybe this is explained in the authors other publication?

Response: Studies have shown that the follicular cell layer is an important synthesis site for hormones, nutrients, and other signaling molecules. In this study, we found that the serum LH/FSH of female fish decreased after heat stress, and the expression of TGFβ1I1 in granulosa cells was also significantly inhibited. As a downstream target gene of TGFβ1, we have reason to believe that TGFβ1I1 is involved in mediating hormone synthesis in the follicular layer. Later, we will further validate the experimental results by injecting miR-33 inhibitors or knocking down the TGFβ1I1 gene.

Comment 3: There is a logical jump between the descriptive work and the sequencing results and the jump from the DE miRNAs to miR-33 is not clear to me. What about the other miRNAs? Could the authors perform some pathway analysis to see whether there are common aspects between the upregulated miRNA and between the downregulated miRNAs? What about 33 vs 34a? Or 27e vs 27b? I appreciate tha the authors found interesting resutls with miR33, but some form of analysis of the remainder of the miRNAs would be well placed here if only to provide a reference for a future paper of the authors looking at another one of these targets.

Response: This study is the first to discover the adverse effects of heat stress on follicular development in Nile tilapia. Through miRNA sequencing to reveal which potential miRNAs are involved in follicle development and atresia under heat stress, this study then focused on the molecular mechanism of miR-33 regulation. Through miRNA sequencing, we found that multiple miRNAs, such as miR-27e, -27b-3p, -33, -34a, -133a-5p and -301b-5p, were differentially expressed. The relationship between different differential miRNAs and their potential functions will be the focus of our next research.

Comment 4: Line 324 - you switch the order in which you express, siognificantly higher and significantly lower meaning the same since you flipped the groups. This could be confusing for the reader and it might be more easily understandable if you keep the order the same in both sentences.

Response: We have revised it [Lines 311-313].

Comment 5: Line 355 - was the qPCR carried out on the same samples as the sequencing or on a different cohort?

Response: Samples for qRT-PCR validation were derived from sequenced samples.

Comment 6: Data for transcript levels is expressed as % of b actin or % of U6. In the methods the authors mention the ddCT method. The results of that method are not in % of control gene. Could you please expand how this data was calculated or correct the legend?

Response: We have corrected the legends of Figure 3, Figure 7 and Figure 8.

Reviewer 2 Report

The present manuscript, entitled “The miR-33/ TGFβ1I1 axis participates in follicular atresia of Nile tilapia (Oreochromis niloticus) by controlling granulosa cell apoptosis” aimed to investigate the effect of heat stress in the tilapia oogenesis. The manuscript is very well written. The objectives proposed for the study were achieved and very well presented in the result section. I address some suggestions for the authors in the topics below:

Title: I strongly suggest for the authors to add some information regarding heat stress in the tittle of the study. I know that the novelty are the effects of miR-33 on fish follicular atresia. However, the applicability of the study resides directly in the effects of the heat on Nile tilapia oogenesis, which is of great importance for aquaculture in general.

Material and Methods

This section should present to the reader the morphologic classification of oocytes used in Fig 2. (Stages II, III, IV and V, and atretic follicle). Moreover, I recommend the use of AF instead of FA for atretic follicle. In the same way, the method used for the follicle quantification should be briefly explained (Table 1).

Results

The true values of P should be presented instead of higher or lower than 0.05.

Figure 3: The function of the red arrows should be presented in the figure subtitle.

Figure 6: The meaning of two asterisk should be explained in the subtitle.

Figure 11: Caspase-3 and Caspase-8. The meaning of two asterisk should be explained in the subtitle.

Author Response

Comment 1: Title: I strongly suggest for the authors to add some information regarding heat stress in the tittle of the study. I know that the novelty are the effects of miR-33 on fish follicular atresia. However, the applicability of the study resides directly in the effects of the heat on Nile tilapia oogenesis, which is of great importance for aquaculture in general.

Response: We have revised the title of manuscript.

Comment 2: This section should present to the reader the morphologic classification of oocytes used in Fig 2. (Stages II, III, IV and V, and atretic follicle). Moreover, I recommend the use of AF instead of FA for atretic follicle. In the same way, the method used for the follicle quantification should be briefly explained (Table 1).

Response: We have added descriptions of oocytes at different stages in Figure 1B. In Table 1, we have added the description and reference for the identification and counting of atretic follicles. In addition, the abbreviation FA for atretic follicles was changed to AF, see Figure 1B.

Comment 3: The true values of P should be presented instead of higher or lower than 0.05.

Response: The manuscript involves a lot of data analysis. It may be too cumbersome to express all of them with P values. It may be clearer to express with P<0.05 or P>0.05.

Comment 4: Figure 3: The function of the red arrows should be presented in the figure subtitle.

Response: We have added the description in Figure 1 and Figure 7.

Comment 5: Figure 6: The meaning of two asterisk should be explained in the subtitle.

Response: We have added the description in Figure 3.

Comment 6: Figure 11: Caspase-3 and Caspase-8. The meaning of two asterisk should be explained in the subtitle.

Response: We have added the description in Figure 8.

Reviewer 3 Report

The paper is well written by Qiang and other collaborators. However, some minor modifications are required as follows.

  1. In the Introduction section of the Ms, kindly add the latest reference in Nile tilapia, PMID: 31664705 and https://doi.org/10.1016/j.aquaculture.2017.11.025
  2. The Materials and Methods section of the Ms is well written.
  3. In figure 1B, the background color must be white.
  4. Figure 7C is not clearly visible. Kindly make it in high-resolution quality.
  5. The maximum number of figures is 8. The rest of the figures can add them in the supplementary section.
  6. Check the grammatical mistake throughout the Ms.

Author Response

Comment 1: In the Introduction section of the Ms, kindly add the latest reference in Nile tilapia, PMID: 31664705 and https://doi.org/10.1016/j.aquaculture.2017.11.025

Response: We have added the reference in the Introduction section.

Comment 2: In figure 1B, the background color must be white

Response: We have revised Figure 1B.

Comment 3: Figure 7C is not clearly visible. Kindly make it in high-resolution quality

Response: We have revised Figure 4C.

Comment 4: The maximum number of figures is 8. The rest of the figures can add them in the supplementary section

Response: We have revised the number of Figures in the manuscript.

Comment 5: Check the grammatical mistake throughout the Ms.

Response: We have carefully revised the sentence of the manuscript based on the similarity report, and grammatical mistake.

Round 2

Reviewer 1 Report

As a general comment - it is really hard to follow changes. Does the journal have guidelines to track changes to the manuscript in red or something in that direction? I found the tracked MS WOrd manuscript - there are fairly far reaching changes to the writing and it looks like I should maybe carefully read the manuscript again but as outlined below I would kindly ask the authors to consider some changes to what I consider duplicate presentation of data.

The authors respond that this paper as well as the Aquaculture are "completely different". I thank the handling editor for sharing the Aquaculture paper - please excuse that my institute had no access.

Figure 1 Genes: Morphology of heat stressed ovaries day 6 and 12
Figure 1 Aquaculture: Morphology of heat stressed ovaries day 0, 2, 4, 8, 16

Figure 2 Genes: H&E histology of heat stressed ovary day 6 and 12
Figure 2 Aquaculture: H&E histology of heat stressed ovary day 0, 2, 4, 8, 16

Figure 3 Genes: TUNEL-stained gonad tissue of heat stressed ovaries day 6 and 12
Figure 3 Aquaculture: TUNEL-stained gonad tissue of heat stressed ovaries day 0, 2, 4, 8, 16

Figure 4 Genes: Serum FSH. LH, E2 of heat stressed ovaries day 6 and 12
Figure 3 Aquaculture: Serum FSH. LH, E2 of heat stressed ovaries day 0, 2, 4, 8, 16

Figure 5 Genes: Volcano plot of differentially expressed genes at 6 and 12 days
Figure 8 Aquaculture: Pathway analysis of differentially expressed genes at 4days after heat stress

The Aquaculture paper goes into differentially expressed genes and regulatory networks.

The Genes paper then goes into miRNA seq and differentailly expressed miRNAs followed by a focus on miR-33.

I agree that the part on miR is new and different. I would however say that a timecourse from 0-16 days includes 6 and 12 days. THe data also look similar, including the serum analysis (E2 is reduced in the heat stressed group from 2, 4, 8, 16 days and now in the new paper we learn that this is also true at 6 and 12 days. Are the authors planning to publish in the next paper that E2 is reduced at 5 and 7 days? In my opinion the authors should remove the duplicated data, reference their Aquaculture paper adequately to express that heat stress leads to altered morphology, increased TUNEL staining and changes in E2, FSH/LH and differentially expressed genes and they present the novel data of this paper as Figure 1 - the miRNA analysis.

From here on, the miR seq data as well as the focus on miR33 and the TGFb western / IHC seem a bit weak for a full paper but I will leave this up to the editor to decide. The authors state that in the next paper they wish to validate miR33 as a target with knockdown of miR33 as well as TGFb1l1 - maybe they already have data to this extent and would like to add them here?

Please present a full western additional to the cropped wastern, maybe in the supplement for the reader to estimate whether there are other genes and where the protein runs with a size marker.

Author Response

Detailed responses to comments from the editors and reviewers are provided below. Changes are shown in the TrackedCopy manuscript.

Reviewer 1:

Comment 1: I agree that the part on miR is new and different. I would however say that a timecourse from 0-16 days includes 6 and 12 days. THe data also look similar, including the serum analysis (E2 is reduced in the heat stressed group from 2, 4, 8, 16 days and now in the new paper we learn that this is also true at 6 and 12 days. Are the authors planning to publish in the next paper that E2 is reduced at 5 and 7 days? In my opinion the authors should remove the duplicated data, reference their Aquaculture paper adequately to express that heat stress leads to altered morphology, increased TUNEL staining and changes in E2, FSH/LH and differentially expressed genes and they present the novel data of this paper as Figure 1 - the miRNA analysis.

From here on, the miR seq data as well as the focus on miR33 and the TGFb western / IHC seem a bit weak for a full paper but I will leave this up to the editor to decide. The authors state that in the next paper they wish to validate miR33 as a target with knockdown of miR33 as well as TGFb1l1 - maybe they already have data to this extent and would like to add them here?

Response: Thank you for your careful review. The presentation of serum hormones, tissue sections and other indicators in this manuscript is to better understand how miR-33 increases follicular atresia by affecting granulosa cell apoptosis. If we only analyze follicular atresia from the molecular level, the content may be relatively poor, and it cannot allow readers to accurately understand the connection between miR-33 and follicular atresia. Also, the contents of serum hormones and tissue sections in this manuscript only account for a small part of the whole study, and the study mainly focuses on the construction, analysis and verification of miRNA library. miR-33 knockout experiments are still ongoing.

Comment 2: Please present a full western additional to the cropped wastern, maybe in the supplement for the reader to estimate whether there are other genes and where the protein runs with a size marker.

Response: We have added the experimental western blot with a size marker to the Supplementary File, see Figure 1S.
